# Omalizumab Restores Response to Corticosteroids in Patients with Eosinophilic Chronic Rhinosinusitis and Severe Asthma

**DOI:** 10.3390/biomedicines9070787

**Published:** 2021-07-07

**Authors:** Yoshiki Kobayashi, Akira Kanda, Dan Van Bui, Yasutaka Yun, Linh Manh Nguyen, Hanh Hong Chu, Akitoshi Mitani, Kensuke Suzuki, Mikiya Asako, Hiroshi Iwai

**Affiliations:** 1Airway Disease Section, Department of Otorhinolaryngology, Kansai Medical University, Hirakata, Osaka 573-1010, Japan; akanda@hirakata.kmu.ac.jp (A.K.); buivanda@hirakata.kmu.ac.jp (D.V.B.); yunys@hirakata.kmu.ac.jp (Y.Y.); nguyenli@hirakata.kmu.ac.jp (L.M.N.); chuh@hirakata.kmu.ac.jp (H.H.C.); mitaniak@hirakata.kmu.ac.jp (A.M.); suzukken@hirakata.kmu.ac.jp (K.S.); asako@takii.kmu.ac.jp (M.A.); iwai@hirakata.kmu.ac.jp (H.I.); 2Allergy Center, Kansai Medical University Hospital, Hirakata, Osaka 573-1010, Japan

**Keywords:** asthma, CCL4, CD69, corticosteroid sensitivity, eosinophilic chronic rhinosinusitis, protein phosphatase 2A

## Abstract

Eosinophilic chronic rhinosinusitis (ECRS), which is a subgroup of chronic rhinosinusitis with nasal polyps, is characterized by eosinophilic airway inflammation extending across both the upper and lower airways. Some severe cases are refractory even after endoscopic sinus surgery, likely because of local steroid insensitivity. Although real-life studies indicate that treatment with omalizumab for severe allergic asthma improves the outcome of coexistent ECRS, the underlying mechanisms of omalizumab in eosinophilic airway inflammation have not been fully elucidated. Twenty-five patients with ECRS and severe asthma who were refractory to conventional treatments and who received omalizumab were evaluated. Nineteen of twenty-five patients were responsive to omalizumab according to physician-assessed global evaluation of treatment effectiveness. In the responders, the levels of peripheral blood eosinophils and fractionated exhaled nitric oxide (a marker of eosinophilic inflammation) and of CCL4 and soluble CD69 (markers of eosinophil activation) were reduced concomitantly with the restoration of corticosteroid sensitivity. Omalizumab restored the eosinophil-peroxidase-mediated PP2A inactivation and steroid insensitivity in BEAS-2B. In addition, the local inflammation simulant model using BEAS-2B cells incubated with diluted serum from each patient confirmed omalizumab’s effects on restoration of corticosteroid sensitivity via PP2A activation; thus, omalizumab could be a promising therapeutic option for refractory eosinophilic airway inflammation with corticosteroid resistance.

## 1. Introduction

Eosinophilic chronic rhinosinusitis (ECRS) is known as a subtype of chronic rhinosinusitis with nasal polyps characterized by type-2-predominant airway inflammation [1,2,3]. Owing to the high incidence of bronchial asthma (80% or more in severe ECRS) [4,5], ECRS is recognized as an intractable eosinophilic airway disease.

Intranasal corticosteroids have limited effectiveness against ECRS. Topical corticosteroids that are administrated intranasally are not delivered to the inflammatory sites, such as the middle meatus, to which the ethmoid sinus opens due to functional–anatomical aspects [6]. Although intermittent oral corticosteroids are effective, they are not suitable for long-term administration owing to their adverse effects. Under such conditions, we have shown the usefulness of inhaled corticosteroid (ICS) exhalation through the nose (ETN) treatment for ECRS patients with bronchial asthma [6,7,8].

Importantly, ICS-ETN treatment reduces the need for surgical interventions, whereas patients with severe ECRS who have a higher nasal polyp score and sinus computed tomography (CT) score require endoscopic sinus surgery (ESS) despite ICS-ETN treatment [6]. Furthermore, approximately 30% of these patients experience recurrence under maintenance therapy with ICS-ETN after ESS. These patients have comorbid bronchial asthma, which is recognized as a risk factor of relapse after ESS [5]. This suggests that severe ECRS with asthma is much more difficult to control, possibly because of reduced corticosteroid sensitivity [9,10].

A randomized, double-blind, placebo-controlled study indicated the clinical efficacy of omalizumab for nasal polyps with comorbid asthma [11]. Local polyclonal IgE in the airway mucosal tissue contributes to local airway inflammation, which could be regulated by omalizumab. Furthermore, treatment with omalizumab for severe allergic asthma also improves the outcome of coexistent ECRS in real-life studies, supporting the concept of ‘one airway, one disease’ [12,13] and the efficacy of omalizumab for intractable eosinophilic airway disease; however, the underlying mechanisms of omalizumab for eosinophilic airway inflammation have not been fully elucidated.

Serine/threonine phosphatase PP2A has been reported to regulate corticosteroid sensitivity via dephosphorylation of the glucocorticoid receptor at Ser^226^ [14]. Furthermore, we recently found that under eosinophilic airway inflammation, such as an ECRS with asthma, impaired PP2A induces local responses to corticosteroids [10].

In this study, we focused on responders to omalizumab in patients with ECRS and asthma and investigated the molecular mechanisms of omalizumab’s action.

## 2. Materials and Methods

### 2.1. Study Subjects

From July 2015 to July 2019, 25 patients with ECRS and severe asthma who were refractory to conventional treatments started to receive omalizumab and were followed-up for more than 1 year at our hospital. ECRS and bronchial asthma were diagnosed according to the Japanese Epidemiological Survey of Refractory Eosinophilic Chronic Rhinosinusitis [5] and the Global Initiative for Asthma guidelines, respectively [15]. All patients had at least 1 positive in vitro reaction to a perennial aeroallergen. Patients were selected according to ‘case pathways’ [16]. Omalizumab (Xolair; Novartis, Basel, Switzerland) was administered subcutaneously every 2 or 4 weeks based on baseline serum total IgE level (30–1500 IU/mL) and body weight. Two patients received omalizumab every 2 weeks and the other 23 patients received omalizumab every 4 weeks according to the official medication leaflet. Changes from baseline in nasal polyp score [17], CT score as defined by the Lund–Mackay scale [18], sense of smell, pulmonary function test, and quality of life score (Asthma Control Test [19]) were assessed. Fractionated exhaled nitric oxide (FENO) was measured as a marker of eosinophilic airway inflammation. Response to corticosteroids was evaluated before and at 4 months after treatment. The clinical response to omalizumab was determined by a physician-assessed global evaluation of treatment effectiveness, using the outcomes described above as a reference. Patients with an ‘excellent’ or ‘good’ response were considered responders, while those with a ‘moderate’, ‘poor’, or ‘worsening’ response were considered non-responders [20]. This study was approved by the local ethics committee of Kansai Medical University (approval number: KanIRin1313).

### 2.2. Cell Preparation

Peripheral blood mononuclear cells (PBMCs) were separated using Ficoll-Paque PLUS^®^ (GE Healthcare, Uppsala, Sweden). Eosinophils (purity > 98%) were isolated from the peripheral blood of healthy volunteers with mild eosinophilia (approximately 4–8% of total white blood cells) by negative selection using a MACS system with an Eosinophil Isolation Kit (Miltenyi Biotec, Bergisch Gladbach, Germany). The human bronchial epithelial cell line BEAS-2B was obtained from the European Collection of Authenticated Cell Culture (Salisbury, UK). BEAS-2B cells were co-incubated with serum obtained from patients for 48 h as needed.

### 2.3. Corticosteroid Sensitivity

PBMCs and BEAS-2B cells were treated with dexamethasone (Sigma-Aldrich, St. Louis, MO, USA) for 45 min, followed by TNFα (10 ng/mL; R&D Systems, Minneapolis, MN, USA) stimulation overnight. The ability of dexamethasone to inhibit TNFα-induced CXCL8 release was determined in cell medium by sandwich ELISA according to the manufacturer’s instructions (R&D Systems). The IC_50_ of dexamethasone on CXCL8 production (Dex-IC_50_), calculated using Prism^®^ 8.0 statistical software (GraphPad, San Diego, CA, USA), was used as a marker for corticosteroid sensitivity.

### 2.4. Quantitative RT-PCR

Total RNA extraction and reverse transcription were performed using a RNeasy Mini Kit (Qiagen, Hilden, Germany) and a PrimeScript RT MasterMix (Perfect Real Time; Takara Bio, Shiga, Japan). Gene transcript levels of protein phosphatase 2 catalytic subunit alpha isozyme (PPP2CA), CCL4, IL-13, inducible nitric oxide synthase (iNOS), and glyceraldehyde 3-phosphate dehydrogenase (GAPDH) were quantified by real-time PCR using a Rotor-Gene SYBR Green PCR kit (Qiagen, Hilden, Germany) on a Rotor-Gene Q HRM (Corbett Research, Cambridge, UK). Appendix A shows details of the amplification primers (Eurofins Genomics, Tokyo, Japan).

### 2.5. Cell Lysis, Immunoprecipitation, and Fluorometric Assay

Cell protein extracts were prepared using modified RIPA buffer (50 mM Tris HCl pH 7.4, 1.0% NP-40, 0.25% Na-deoxycholate, 150 mM NaCl with freshly added complete protease inhibitor). Protein concentrations were determined using the BCA Protein Assay (Thermo Fisher Scientific, Rockford, IL, USA). Immunoprecipitation was conducted with anti-PP2A antibody (Santa Cruz Biotechnology, Dallas, TX, USA). Phosphatase activity in immunopurified PP2A was assayed using SensoLyte^TM^ MFP Protein Phosphatase Assay Kit (AnaSpec, San Jose, CA, USA), as previously described [14]. The activity of DNAse I in cell extracts was assayed using the DNAse I Activity Assay kit (Bio Vision, Milpitas, CA, USA).

### 2.6. Immunoassay for CCL4, IL-13, and CD69

CCL4 and IL-13 levels in serum or cell culture supernatant were measured using MIP-1β/CCL4 and IL-13 ELISA kits (R&D Systems). Purified eosinophils were incubated with BEAS-2B cells overnight. Monoclonal antibodies against Siglec-8 (APC-conjugated) and CD69 (BV610-conjugated) (BioLegend, San Jose, CA, USA) were added to eosinophils fixed by 4% paraformaldehyde. Then, 30 min after reaction, the CD69 expression levels on eosinophils further identified by Siglec-8 were measured using a flow cytometer (FCM) and determined relative to the expression level of the isotypic control. In addition, soluble levels in serum were assayed using the soluble CD69 ELISA kit (Aviscera Bioscience, Inc., Santa Clare, CA, USA).

### 2.7. RNA Interference

PP2ACα siRNAs and non-silencing scrambled control siRNA were purchased from Qiagen. The siRNA sequences (0.6 μM) were transfected using Lipofectamine RNAiMAX Reagent (Invitrogen, Carlsbad, CA, USA) according to the manufacturer’s specifications.

### 2.8. Immunofluorescence Staining

After co-incubation with purified eosinophils overnight, BEAS-2B cells were fixed with 4% formaldehyde for 20 min, permeabilized, and blocked. The cells were then incubated with the rabbit polyclonal antibody to PP2A (GeneTex, Alton Pkwy Irvine, CA, USA), followed by APC-labeled donkey anti-rabbit antibody (Jackson Immuno Research, West Grove, PA, USA). Control antibodies and Hoechst (Invitrogen, Paisley, UK) were included in each experiment. PP2A expression was evaluated using a Carl Zeiss LSM700 confocal microscope.

### 2.9. Cell Survival

The viability of purified eosinophils was evaluated using double staining with annexin V and 7-amino-actiomycin D (7-AAD) (BD Pharmingen, Franklin Lakes, NJ, USA).

### 2.10. Viscoelasticity

Mucin samples obtained from the sinuses of ECRS patients were cut into uniformly sized pieces and resuspended in culture media. The viscoelasticity of each mucin sample was measured using a viscometer (LVDV2PCP, EKO Instruments, Co. Ltd., Tokyo, Japan) with a spindle (CPA-52Z, EKO Instruments).

### 2.11. Statistical Analysis

Comparisons of two groups of data were performed using the Mann–Whitney U test or paired t-test. Other data were analyzed using ANOVA with a post hoc test adjusted for multiple comparisons, as appropriate. Differences were considered statistically significant for *p* values < 0.05. Descriptive statistics were expressed as means ± SEM.

## 3. Results

### 3.1. In Responders to Omalizumab, Blood Eosinophils and FENO Are Reduced with Restoration of Response to Corticosteroids

Nineteen out of twenty-five patients were responsive to omalizumab. We found no significant differences in baseline characteristics, except for complications of eosinophilic otitis media between the responders and the non-responders (Table 1). Peripheral blood eosinophils and FENO were reduced 4 months after treatment with omalizumab in the responders (Figure 1A,B) in parallel with increases in forced expiratory volume in 1 s (FEV_1_) and ACT, a reduction of nasal polyp size, and opacification of paranasal sinus CT (Appendix A). Serum levels of CCL4, a chemokine for inflammatory cells including eosinophils [21]; and serum-soluble CD69, a marker for eosinophil activation [22], were also reduced in responders (Figure 1C,D). In addition, corticosteroid sensitivity in PBMCs obtained from the responders increased by 4 months of omalizumab treatment (Figure 1E).

### 3.2. PP2A Is Associated with Eosinophilic Airway Inflammation

To examine the mechanisms of omalizumab’s effects, we examined the association between the clinical markers of eosinophilic airway inflammation, such as CCL4, CD69, and IL-13, an inducer of iNOS [23] and PP2A (which regulates corticosteroid sensitivity via nuclear translocation of the glucocorticoid receptor), in BEAS-2B airway epithelial cells. Co-incubation with purified eosinophils and eosinophil peroxidase, an eosinophil granule protein, increased CCL4 mRNA expression and release (Figure 2A). Interestingly, CCL4 enhanced CD69 expression on eosinophils (Figure 2B and Appendix A), which suggests that CCL4 might be involved in eosinophil activation. IL-13 mRNA expression and release were also increased by co-incubation with purified eosinophils and eosinophil peroxidase, concomitantly with the elevation of iNOS mRNA expression (Figure 2C,D). More importantly, co-incubation with purified eosinophils and eosinophil peroxidase reduced PP2A expression, while PP2A reduction by siRNA enhanced CCL4, IL-13, and iNOS expression (Figure 2E,F). These findings suggest that PP2A might be a therapeutic target of omalizumab for eosinophilic airway inflammation.

### 3.3. Omalizumab Restores PP2A Activity and Corticosteroid Sensitivity in Airway Epithelial Cells

To confirm the effects of omalizumab on corticosteroid sensitivity via PP2A activation, we focused on the PP2A activity in airway epithelial cells. Eosinophil peroxidase reduced the PP2A activity and responsiveness to corticosteroids in BEAS-2B bronchial epithelial cells, whereas omalizumab restored the PP2A activity and corticosteroid sensitivity (Figure 3A,B) and reduced CCL4, IL-13, and iNOS expression (Appendix A). Next, in order to reproduce the airway inflammation of each patient, BEAS-2B was incubated with diluted serum from each patient. In responders to omalizumab, corticosteroid sensitivity in this local inflammation simulant model improved with 4 months of omalizumab treatment concomitantly with the restoration of PP2A activity (Figure 3C,D). CCL4 and IL-13 release from BEAS-2B in the same model were also reduced in the responders (Figure 3E,F).

### 3.4. Omalizumab Promotes Mucin Decomposition

Furthermore, to examine the direct effects of omalizumab on eosinophilic inflammation, we focused on the effects of omalizumab on the viability of eosinophils, CD69 expression on eosinophils, and mucin decomposition. Omalizumab did not induce eosinophil apoptosis, which was reduced by co-incubation with BEAS-2B (Figure 4A). In addition, omalizumab did not inhibit CD69 expression on eosinophils, which was enhanced by co-incubation with BEAS-2B (Figure 4B). This suggests that omalizumab indirectly regulates eosinophilic inflammation via the restoration of corticosteroid sensitivity. On the other hand, omalizumab restored the DNAase I activity in BEAS-2B, which was reduced by co-incubation with eosinophils (Figure 4C). As a result, the mucin decomposition inhibited under co-incubation with BEAS-2B was promoted by treatment with omalizumab (Figure 4D).

## 4. Discussion

To clarify the underlying mechanisms of how omalizumab regulates eosinophilic airway inflammation, we examined the differences between responders and non-responders to omalizumab in patients with ECRS and severe asthma who were refractory to conventional treatments. In responders, corticosteroid sensitivity was restored concomitant with a reduction of peripheral blood eosinophils and FENO.

As one of the mechanisms of omalizumab’s effect on eosinophilic airway inflammation, although omalizumab did not exert a direct effect on eosinophil apoptosis in our in vitro experiment, it induced eosinophil apoptosis in vivo [24], which supports our findings that omalizumab reduced not only peripheral blood eosinophil counts, but also nasal polyps and sinus opacification with eosinophil-dominant infiltration. Another report showed that omalizumab reduced FcεRI expression on circulating dendritic cells, which might lead to a reduction in allergen presentation and Th2 cell activation and proliferation [25]. Th2 cytokines (such as interleukin (IL)-4, IL-5, and IL-13) are produced by CD4^+^ T cells and are believed to play a key role in asthma pathogenesis by promoting recruitment and activation of mast cells and eosinophils, which are the primary effector cells in the allergic response [26]. It has been reported that omalizumab reduces airway infiltration of CD4^+^ T cells, as well as eosinophils and FcεRI expression cells [27], which might be associated with the CCL4 reduction observed in this study. CCL4 is known as a chemoattractant against eosinophils and immune cells (e.g., monocytes and T cells) and is produced by inflammatory cells, including activated eosinophils, immune cells, and epithelial cells [21,28]. Furthermore, CCL4 might be involved in eosinophil activation, as indicated by CCL4-induced CD69 expression. We confirmed lower CCL4 release in the local inflammation simulant model after omalizumab treatment; thus, suppression of CCL4 could lead to control of eosinophilic airway inflammation.

We recently reported that eosinophil peroxidase, an eosinophil granule protein, reduced PP2A activity via induction of its phosphorylation, resulting in corticosteroid insensitivity [10]. Similarly, in this study, we found that PP2A expression was reduced under eosinophilic inflammation, which led to enhancement of CCL4, IL-13, and iNOS expression, indicators of type 2 inflammation. Further, in BEAS-2B cells treated with PP2A siRNA, responses to corticosteroids in the culture media were reduced, resulting in the enhancement of these indicators. In responders to omalizumab, reduction of eosinophils at inflammatory sites by omalizumab treatment could prevent inactivation of PP2A indirectly via less eosinophil peroxidase release. On the other hand, we found that omalizumab could directly activate PP2A, although we still need to clarify the mechanism. Upregulation of PP2A activity by omalizumab treatment could restore corticosteroid sensitivity and improve the efficacy of topical corticosteroid therapy, concomitantly with reduction of indicators of type 2 inflammation.

In this study, we also focused on serum-soluble CD69. CD69 expression on peripheral blood eosinophils was higher in patients with ECRS than in patients with chronic rhinosinusitis (non-ECRS) [29]. Importantly, in the same patients with ECRS, CD69 expression on eosinophils was extremely elevated in nasal polyps compared with that in peripheral blood. Furthermore, CD69 expression on eosinophils was significantly enhanced by co-incubation with BEAS-2B cells or CCL4, which suggests that CD69 could be an eosinophil activation marker, at least at the eosinophilic inflammatory site. A previous report showed that CD69+ peripheral blood eosinophils in patients with allergic asthma were not reduced after treatment with omalizumab [24], consistent with our in vitro experiment, in which omalizumab did not inhibit CD69 expression on eosinophils; however, we confirmed that serum-soluble CD69 in the patients with ECRS and severe asthma was reduced in the responders to omalizumab treatment, which indicates that soluble CD69 could reflect the activity of eosinophilic inflammation in severe cases.

Notably, we found that omalizumab exerts another direct effect for eosinophilic inflammation. Omalizumab restored the reduced DNAse I activity in airway epithelial cells under eosinophilic inflammation and promoted mucin decomposition, which contributes to the improvement of the delivery of topical corticosteroids.

On the contrary, the non-responders had peripheral blood eosinophilia (peak levels were 1000/μL or more) or complications of eosinophilic otitis media (EOM). EOM is a refractory otitis media characterized by accumulation of eosinophils in the middle ear with eosinophilic mucin, and is highly associated with ECRS and asthma [30,31]. Mepolizumab or dupilumab has been reported to be effective for refractory EOM [32,33], and the non-responders in this study responded to these molecular targeted therapies, suggesting that patients with EOM have more severe local type 2-dominant eosinophilic inflammation and might be less responsive to omalizumab.

## 5. Conclusions

Taken together, omalizumab has the potential to inhibit eosinophil activation and infiltration into the local inflammatory site owing to restoration of corticosteroid sensitivity through PP2A activation. Omalizumab could be a promising therapeutic option in refractory eosinophilic airway inflammation with corticosteroid resistance.

## Figures and Tables

**Figure 1 biomedicines-09-00787-f001:**
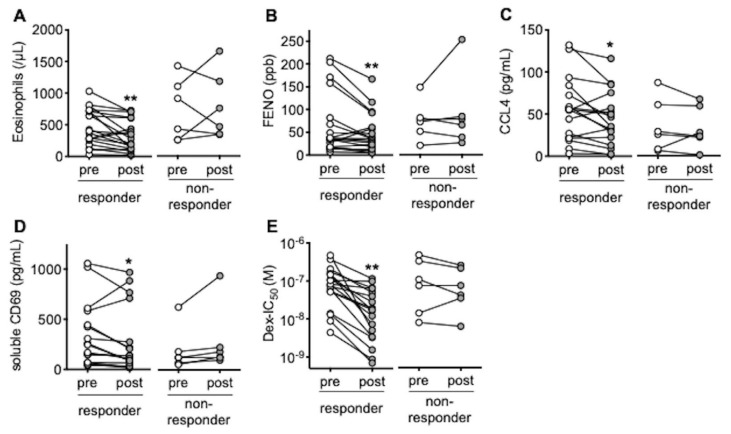
Effects of omalizumab on clinical markers and corticosteroid sensitivity. Peripheral blood eosinophil counts (**A**), FENO (**B**), serum CCL4 levels (**C**), serum-soluble CD69 levels (**D**), and IC_50_ values for dexamethasone on TNFα-induced CXCL8 production (Dex-IC_50_) in PBMCs, a marker of corticosteroid sensitivity (**E**), were evaluated pre- and post-treatment with omalizumab. Individual values of the responder group (*n* = 19) and the non-responder group (*n* = 6) are shown; * *p* < 0.05, ** *p* < 0.01 (vs. pre-treatment).

**Figure 2 biomedicines-09-00787-f002:**
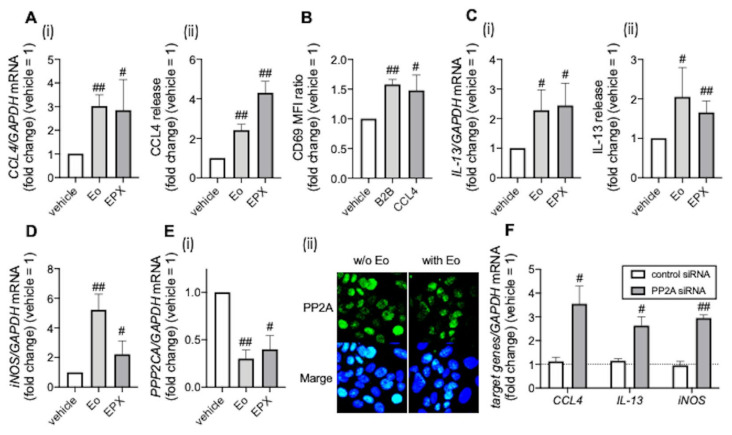
Association between clinical markers of eosinophilic airway inflammation and PP2A. (**A**,**C**–**E**) BEAS-2B cells were co-incubated with purified peripheral blood eosinophils overnight or recombinant eosinophil peroxidase (EPX, 10 μg/mL; Proteintech, Rosemont, IL, USA) for 72 h. CCL4 (**A**) and IL-13 expression levels (**C**) ((i) mRNA levels, (ii) protein levels in culture supernatants), iNOS mRNA levels (**D**), and PP2A expression (**E**) ((i) mRNA levels, (ii) protein levels) were evaluated. (**B**) Purified peripheral blood eosinophils were incubated with BEAS-2B cells or CCL4 (10 μg/mL; Abcam, Cambridge, UK) overnight. CD69 expression on eosinophils was evaluated. (**F**) mRNA levels of CCL4, IL-13, and iNOS in BEAS-2B cells treated with siRNA were evaluated. Values in panels represent the means ± SEM values of four (**A**–**E**) or three experiments (**F**); ^#^
*p* < 0.05, ^##^
*p* < 0.01 (vs. vehicle or control siRNA). (ii) Images were obtained using a Carl Zeiss LSM700 confocal microscope (400 × objective). Results are representative of at least four experiments.

**Figure 3 biomedicines-09-00787-f003:**
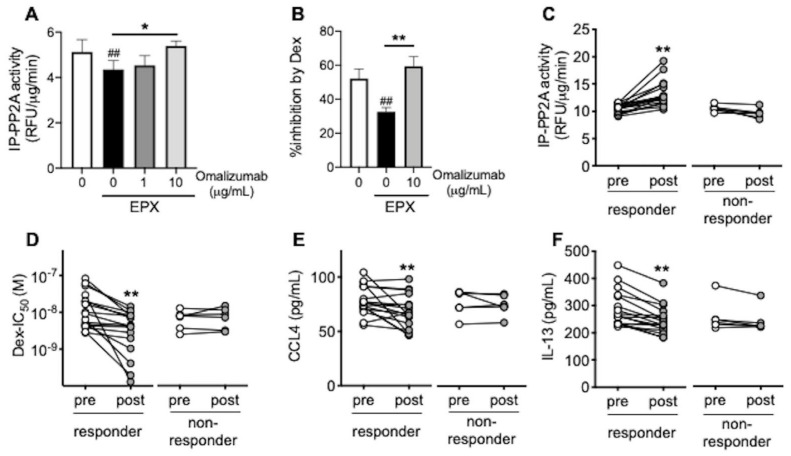
Effects of omalizumab on PP2A activity and corticosteroid sensitivity in airway epithelial cells. (**A**,**B**) BEAS-2B cells co-incubated with recombinant eosinophil peroxidase (EPX, 10 μg/mL) for 72 h were treated with omalizumab (1 or 10 μg/mL) for 20 min. (**C**–**E**) BEAS-2B cells were incubated with 1:10-diluted serum from each patient (pre- and post-treatment with omalizumab) for 48 h. Phosphatase activity in immunoprecipitates with PP2A (IP-PP2A) (**A**,**C**), % inhibition by dexamethasone (Dex) on TNFα-induced CXCL8 production (**B**), and IC_50_ values for dexamethasone (Dex-IC_50_) (**D**) on TNFα-induced CXCL8 production and CCL4 and IL-13 levels in supernatants (**E**,**F**) were evaluated. (**A**,**B**) Values represent the means ± SEM values of four experiments; ^##^
*p* < 0.01 (vs. non-treatment control), * *p* < 0.05, ** *p* < 0.01 (as shown between the two groups). (**C**–**F**) Individual values in the responder group (*n* = 19) and the non-responder group (*n* = 6) are shown; ** *p* < 0.01 (vs. pre-treatment).

**Figure 4 biomedicines-09-00787-f004:**
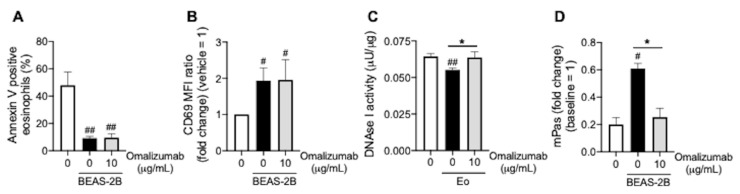
Effects of omalizumab on the DNAse I activity in airway epithelial cells. (**A**,**B**) Purified peripheral blood eosinophils co-incubated with BEAS-2B cells were treated with omalizumab (10 μg/mL) overnight. The proportions of Annexin-V-positive eosinophils (**A**) and CD69 expression on eosinophils (**B**) were evaluated. (**C**) BEAS-2B cells co-incubated with purified peripheral blood eosinophils were treated with omalizumab (10 μg/mL) overnight. The DNAse I activity in BEAS-2B cell extracts was assayed. (**D**) Mucin co-incubated with BEAS-2B was treated with omalizumab (10 μg/mL) overnight. The mucin viscoelasticity was measured using a viscometer at 10 rpm. Values represent the means ± SEM values of four experiments; ^#^
*p* < 0.05, ^##^
*p* < 0.01 (vs. non-treatment control), * *p* < 0.05 (as shown between the two groups).

**Table 1 biomedicines-09-00787-t001:** Patients’ baseline characteristics.

	Responder (*n* = 19)	Non-Responder (*n* = 6)
Age	52.1 ± 13.1	54.0 ± 15.1
Gender (M/F)	8/11	2/4
Body mass index	23.6 ± 3.8	21.3 ± 3.6
NSAIDs intolerance	8	2
EOM *	4	5
Smoking history (never/ex)	13/6	4/2
ESS history (Y/N)	17/2	6/0
Total IgE (IU/mL)	429 ± 403	401 ± 275
Positive RAST (single/multi)	2/17	1/5
Eosinophils (/μL) [peak value]	466 ± 287 [699 ± 328]	741 ± 495 [1022 ± 457]
FENO (ppb)	65.2 ± 67.6	75.8 ± 42.4
Lund-Mackay scale	16.0 ± 4.5	15.3 ± 4.8
Polyp score	4.6 ± 1.6	4.7 ± 2.0
JESREC score	15.1 ± 1.7	16.7 ± 0.8
Impaired sense of smell	13	5
FEV_1_%pred.	82.6 ± 18.2	77.6 ± 22.7
FEF_25–75_%pred.	55.5 ± 30.4	48.3 ± 25.9
FVC %pred.	93.9 ± 15.8	96.0 ± 11.8
Asthma Control Test	21.3 ± 3.3	20.7 ± 5.1
Asthma exacerbation (per year) **	1.4 ± 1.5	1.2 ± 0.4
**Treatment**		
Inhaled corticosteroids (μg) ***	1200 ± 330	1133 ± 350
LABA	19	6
LAMA	5	1
LTRA	16	4
Theophylline	3	1
Anti-histamine	7	3
Inhaled nasal corticosteroids	12	3
Oral corticosteroids	2	0
Omalizumab (mg) (per month)	426 ± 251	425 ± 148

EOM = eosinophilic otitis media; ESS = endoscopic sinus surgery; FENO = fractionated exhaled nitrogen oxide; FEV_1_ = forced expiratory volume in 1 s; FEF_25–75_ = forced expiratory flow between 25% and 75% of vital capacity; FVC = forced vital capacity; JESREC = Japanese Epidemiological Survey of Refractory Eosinophilic Chronic Rhinosinusitis; LABA = long-acting β_2_-agonist; LAMA = long-acting muscarinic antagonist; LTRA = leukotriene receptor antagonist; RAST = radioallergosorbent test’ * *p* < 0.05 (Fisher’s exact test); ** clinically significant exacerbation, defined as any worsening of asthma considered by the treating physician to require systemic corticosteroids; *** equivalent doses of fluticasone propionate. Values are number of subjects and mean ± standard deviation.

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
