# Peer review of "Omalizumab Restores Response to Corticosteroids in Patients with Eosinophilic Chronic Rhinosinusitis and Severe Asthma"

_biomedicines, 2021, doi:10.3390/biomedicines9070787_

Round 1
Reviewer 1 Report
The authors provide a very interesting report showing omalizumab treatment mechanism of action at improvement of severe asthma and ECRS, reporting its role in recovery of corticosteroid sensitivity by restoring PP2A and controlling at some extent eosinophil proinflammatory role. The study is really innovative, well planned, very well performed and explained. Therefore, I suggest some minor revisions, and addition of minor experimental or previously published data to better wrap all the results from the authors to make the research top notch in the field.
- In lines 48-49 please describe briefly omalizumab mechanism of action.
- In line 69 you state that “Omalizumab was administered subcutaneously every 2–4 weeks based on baseline serum total immunoglobulin E level and body weight”, could you be more specific about the cutoff levels of IgE for administration of omalizumab?
- In line 97-98 there is a repetition of the sentence “San Diego, CA), was used as a marker for corticosteroid sensitivity”.
- In figure 1 authors show that PBMCs from responders are more sensitive to dex. Do you have any information about if this restoration also comes accompanied with augmented PP2A expression or activity in these cells?
- In figure 2 it would be nice to include a bar in each graph for the vehicle treated control, for more easier observation of intergroup differences.
- The authors show that omalizumab restores PP2A activity of BEAS2B, and show that this can be related to better dex anti-inflammatory effect, but a direct relationship is not shown. Maybe it would be nice to analyse how dex affects IL13, iNOS and CCL4 in cells treated with PP2A siRNA, or at least provide a clear reference that relates both processes.
- Besides this, in figure 2F authors show that PP2A inhibition alone without Dex treatment augments IL13, iNOS and CCL4 expression. Please in the discussion add some information about how this direct effect of PP2A is performed over cytokine synthesis independent on exogenous corticosteroid treatment process.
- For the protein levels of PP2A in BEAS2B cells can you provide a quantitative measurement of the reduction of PP2A by Eosinophils coculture observed by microscopy? Also, have you done microscopic imaging of BEAS2B cells treated with EPX as in the rest of experiments?
- In figure 3B please describe in more detail what the % of inhibition represents, as I assume it is related to the reduction of TNFα, but it needs to be better explained, as this measurement is only done in these experiments.
- In the conclusion authors suggest that CCL4 reduction by omalizumab is responsible for the restore of glucocorticoid activity, but from the results this is not fully confirmed, as other factors such as IL13 are also deregulated by the serum of the responders. Therefore, conclusion should be rephrased to a more concrete and result-related conclusion as in the abstract.
- Revise the references in the text, they should be before the full stop dots, as in other publications by the journal.
Author Response
Reviewer #1:
We would like to thank for Reviewer’s favorable comments. We tried to answer the questions.
Comments:
- In lines 48-49 please describe briefly omalizumab mechanism of action.
(Response)
According to reviwer’s suggestion, we briefly mentioned about it as follows;
“Local polyclonal IgE in the airway mucosal tissue contributes to local airway inflammation, which could be regulated by omalizumab.”
- In line 69 you state that “Omalizumab was administered subcutaneously every 2–4 weeks based on baseline serum total immunoglobulin E level and body weight”, could you be more specific about the cutoff levels of IgE for administration of omalizumab?
(Response)
As reviewer suggested, we added some information as follows;
“Omalizumab (Xolair; Novartis, Basel, Switzerland) was administered subcutaneously every 2 or 4 weeks based on baseline serum total IgIE level (30–1,500 IU/mL) and body weight. Two patients were received omalizumab every 2 weeks and the other 23 patients were received omalizumab every 4 weeks according to the official medication leaflet.”
- In line 97-98 there is a repetition of the sentence “San Diego, CA), was used as a marker for corticosteroid sensitivity”.
(Response)
We removed it.
- In figure 1 authors show that PBMCs from responders are more sensitive to dex. Do you have any information about if this restoration also comes accompanied with augmented PP2A expression or activity in these cells?
(Response)
Unfortunately, we were not able to evaluate PP2A activity in PBMCs after treatment with omalizumab for the lack of blood sample.
- In figure 2 it would be nice to include a bar in each graph for the vehicle treated control, for more easier observation of intergroup differences.
(Response)
As reviewer suggested, we added the bars of vehicle treated control in Figure 2 A, B, C, D and E (i) .
- The authors show that omalizumab restores PP2A activity of BEAS2B, and show that this can be related to better dex anti-inflammatory effect, but a direct relationship is not shown. Maybe it would be nice to analyse how dex affects IL13, iNOS and CCL4 in cells treated with PP2A siRNA, or at least provide a clear reference that relates both processes.
(Response)
Although we did not perform it in cells treated with PP2A siRNA, we examined the effect of omalizumab-mediated PP2A activation on IL-13, iNOS and CCL4 expression in cells treated with EPX, which induces PP2A reduction. As a result, omalizumab reduced IL-13, iNOS and CCL4, indicating that omalizumab exerted anti-inflammtory effect via PP2A activation. We added these findings as Supplemental Figure S3 and some information in the Result 3.3. and Discussion section.
- Besides this, in figure 2F authors show that PP2A inhibition alone without Dex treatment augments IL13, iNOS and CCL4 expression. Please in the discussion add some information about how this direct effect of PP2A is performed over cytokine synthesis independent on exogenous corticosteroid treatment process.
(Response)
As BEAS-2B cells were cultured with media containing corticosteroids, we found that responses to corticosterioids were reduced in the cells treated with PP2A siRNA, resulting in enhancement of CCL4, IL-13 and iNOS expression. We mentioned it in the Discussion section as follows;
“Further, in BEAS-2B cells treated with PP2A siRNA, responses to corticosteroids in the cultre media were reduced, resulting in the enhancement of these indicators.”
- For the protein levels of PP2A in BEAS2B cells can you provide a quantitative measurement of the reduction of PP2A by Eosinophils coculture observed by microscopy? Also, have you done microscopic imaging of BEAS2B cells treated with EPX as in the rest of experiments?
(Response)
Regarding microscopic imaging of BEAS-2B co-cultured with eosinophils, results were representative of at least four experiments. We added this information to Figure legend. However, we did not a quantitative measurement of microscopic imaging.
We have recently showed the reduction of PP2A protein levels in BEAS-2B treated with EPX using western blotting[1]. Therefore, we did not do microscopic imaging of BEAS-2B treated with EPX.
- In figure 3B please describe in more detail what the % of inhibition represents, as I assume it is related to the reduction of TNFα, but it needs to be better explained, as this measurement is only done in these experiments.
(Response)
As reviewer suggested, we added the explanation in Figure legend as follows;
“% inhibition by dexamethasone (Dex) on TNFa-induced CXCL8 production.”
- In the conclusion authors suggest that CCL4 reduction by omalizumab is responsible for the restore of glucocorticoid activity, but from the results this is not fully confirmed, as other factors such as IL13 are also deregulated by the serum of the responders. Therefore, conclusion should be rephrased to a more concrete and result-related conclusion as in the abstract.
(Response)
According to reviewer’s suggestion, we rephrased conclusion as follows;
“Omalizumab has the potential to inhibit eosinophil activation and infiltration into the local inflammatory site owing to restoration of corticosteroid sensitivity through PP2A activation.”
- Revise the references in the text, they should be before the full stop dots, as in other publications by the journal.
(Response)
As reviewer suggested, we corrected them.
Refferences
- Kobayashi, Y., A. Kanda, Y. Yun, B. Dan Van, K. Suzuki, S. Sawada, M. Asako, and H. Iwai. "Reduced Local Response to Corticosteroids in Eosinophilic Chronic Rhinosinusitis with Asthma." Biomolecules 10, no. 2 (2020).
Reviewer 2 Report
The paper entitled „Omalizumab restores response to corticosteroids in patients with eosinophilic chronic rhinosinusitis and severe asthma” deals with the problem of mechanisms of action of omalizumab in patients insensitive to corticosteroids. Up to now, such mechanisms in eosinophilic chronic rhinosinusitis (ECRS) have not been known, and the paper brings light to the problem. The authors explain the mode of action of the compound on cellular level in a clear way. Only the discussion is rather short and omits the influence of Omalizumab treatment on patients’ further treatment and life. Apart of that, the paper is well written, and easy to follow.
However, I would have some questions to the authors and maybe you could consider to include these issues into an expanded discussion.
- Could you please explain why in ECRS intranasal corticosteroids do not work? Is there a specific mechanism other than corticosteroids resistance or is it just a question of mechanics and maybe a too high speed of the inhaled air during nasal inhalation together with some physical properties of the drug carrier that lowers the time of contact of the drug with the mucosa as you point that inhaled corticosteroids do work when exhaled via the nose?
- In Table 1 you present the patients’ baseline characteristics dividing them to responders and non-responders to Omalizumab. Some of them did not tolerate NDSIDs (more responders) and had a impaired sense of smell (more non-responders). Is there any relationship with the outcome of the study or is it just a coincidence?
- In the introduction you mention that inhaled nasal corticosteroids are not effective in ECRS. At the beginning of your study (Table 1) more than half of the patients used them. Do you have an idea why they received this (probably non-working) treatment?
- Did the treatment with Omalizumab change the patients’ regular treatment and ECRS/asthma outcomes like use of LABA/LAMA/LTRA, lung parameters, quality of life?
- What was the reason (if there was any) that some of the patients did not respond to Omalizumab?
Please check all the names and symbols of all reagents you used in the study (HCL instead of HCl) and provide the location of all manofacturers.
Author Response
Reviewer #2:
We would like to thank for Reviewer’s favorable comments. We tried to answer the questions.
Comments:
- Could you please explain why in ECRS intranasal corticosteroids do not work? Is there a specific mechanism other than corticosteroids resistance or is it just a question of mechanics and maybe a too high speed of the inhaled air during nasal inhalation together with some physical properties of the drug carrier that lowers the time of contact of the drug with the mucosa as you point that inhaled corticosteroids do work when exhaled via the nose?
(Response)
As we indicated in our previous report[1], tropical corticosteroids which are administrated intranasally are not delivered to the inflammatory sites, such as the middle meatus to which the ethmoid sinus opens due to functional-anatomical aspect. We added this information to the introduction section.
- In Table 1 you present the patients’ baseline characteristics dividing them to responders and non-responders to Omalizumab. Some of them did not tolerate NDSIDs (more responders) and had a impaired sense of smell (more non-responders). Is there any relationship with the outcome of the study or is it just a coincidence?
(Response)
Regarding these points, at least there were not significant differences. We think that it is a coincidence.
- In the introduction you mention that inhaled nasal corticosteroids are not effective in ECRS. At the beginning of your study (Table 1) more than half of the patients used them. Do you have an idea why they received this (probably non-working) treatment?
(Response)
As this sentense is a misleading expression, we replaced it as follows;
“Intranasal corticosteroids have limited effectiveness against ECRS.”
- Did the treatment with Omalizumab change the patients’ regular treatment and ECRS/asthma outcomes like use of LABA/LAMA/LTRA, lung parameters, quality of life?
(Response)
Although the treatment with omalizumab did not change regular treatment, it improved pulmonary function test (FEV1), quality of life score (Asthma Control Test), nasal polyp and sinus CT score as indicated in Supplemental Figure S1.
- What was the reason (if there was any) that some of the patients did not respond to Omalizumab?
(Response)
The non-responders had peripheral blood eosinophilia (peak levels were 1,000/mL or more) and/or complication of eosinophilic otitis media (EOM). EOM is a refractory otitis media characterized by accumulation of eosinophils in the middle ear with eosinophilic mucin and is highly associated with ECRS and asthma [2, 3]. Mepolizumab or dupilumab has been reported to be effective for refractory EOM[4, 5] and the non-responders in this study responded to these molecular targeted therapies, suggesting that patients with EOM have more severe local type 2-dominant eosinophilic inflammation and might be less responsive to omalizumab. We mentioned about it in the Discussion section.
- Please check all the names and symbols of all reagents you used in the study (HCL instead of HCl) and provide the location of all manufacturers.
(Response)
As reviewer suggested, we corrected it and added some information in the Material and Methods section and the Figure legends section.
Refferences
- Kobayashi, Y., H. Yasuba, M. Asako, T. Yamamoto, H. Takano, K. Tomoda, A. Kanda, and H. Iwai. "Hfa-Bdp Metered-Dose Inhaler Exhaled through the Nose Improves Eosinophilic Chronic Rhinosinusitis with Bronchial Asthma: A Blinded, Placebo-Controlled Study." Front Immunol 9 (2018): 2192.
- Seo, Y., M. Nonaka, E. Tagaya, J. Tamaoki, and T. Yoshihara. "Eosinophilic Otitis Media Is Associated with Asthma Severity and Smoking History." ORL J Otorhinolaryngol Relat Spec 77, no. 1 (2015): 1-9.
- Ueki, S., N. Ohta, M. Takeda, Y. Konno, and M. Hirokawa. "Eosinophilic Otitis Media: The Aftermath of Eosinophil Extracellular Trap Cell Death." Curr Allergy Asthma Rep 17, no. 5 (2017): 33.
- Iino, Y., E. Takahashi, S. Ida, and S. Kikuchi. "Clinical Efficacy of Anti-Il-5 Monoclonal Antibody Mepolizumab in the Treatment of Eosinophilic Otitis Media." Auris Nasus Larynx 46, no. 2 (2019): 196-203.
- Iino, Y., Y. Sekine, S. Yoshida, and S. Kikuchi. "Dupilumab Therapy for Patients with Refractory Eosinophilic Otitis Media Associated with Bronchial Asthma." Auris Nasus Larynx 48, no. 3 (2021): 353-60.
Round 2
Reviewer 2 Report
Dear Authors,
Thank you very much for your explanations. Now, everything is clear. Just add a 3rd star in line 207, it got lost somewhere.
This manuscript is a resubmission of an earlier submission. The following is a list of the peer review reports and author responses from that submission.